

# Smart Parking

## System do zarządzania smart parkingami online

**Autorzy**: Kacper Dusza◉ · Oskar Gorgis◉ · Karolina Łukasik◉ · Jakub Szmidla◉

**Opiekun:** Martin Tabakow

### Streszczenie

Smart Parking to innowacyjny system zarządzania parkingami, mający na celu usprawnienie procesu parkowania w miastach. Celem projektu jest opracowanie kompleksowego rozwiązania, które umożliwi kierowcom efektywne wyszukiwanie wolnych miejsc parkingowych w najbliższej okolicy, jednocześnie wspierając właścicieli parkingów i władze miejskie w optymalizacji zarządzania przestrzenią parkingową. System został wzbogacony o zaawansowane rozwiązanie do detekcji liczby wolnych miejsc, wykorzystujące techniki sztucznej inteligencji. W wyniku prac powstał system detekcji dostępności miejsc, intuicyjna aplikacja mobilna dla kierowców oraz funkcjonalna aplikacja webowa dla właścicieli parkingów. Znaczenie projektu opiera się na poprawie płynności ruchu miejskiego, oszczędności czasu kierowców, redukcji emisji spalin poprzez ograniczenie niepotrzebnego krążenia pojazdów oraz dostarczeniu narzędzi ułatwiających codzienną mobilność w mieście. Projekt ten stanowi znaczący krok w kierunku inteligentnych miast przyszłości, łącząc korzyści biznesowe z postępem technologicznym i troską o środowisko.

## 1 WSTĘP

Współczesne metropolie zmagają się z wieloma wyzwaniami związanymi z zarządzaniem ruchem miejskim i ograniczeniem emisji spalin. Jednym z istotnych problemów jest nadmierne poszukiwanie miejsc parkingowych, generując przy tym dodatkowy ruch, hałas i emisję $CO_2$. Ponadto brak zintegrowanego systemu informacji o dostępnych miejscach parkingowych utrudnia kierowcom optymalny wybór tras i parkingów.

Właściciele parkingów, którzy nie są w stanie efektywnie zarządzać swoją infrastrukturą lub sukcesywnie reklamować swoją działalność prowadzą do niewykorzystanego potencjału dostępnej przestrzeni parkingowej. Może to znacznie wpłynąć na rentowność małych biznesów lub spowodować słabe wykorzystanie całej sieci parkingów dużych instytucji miejskich.

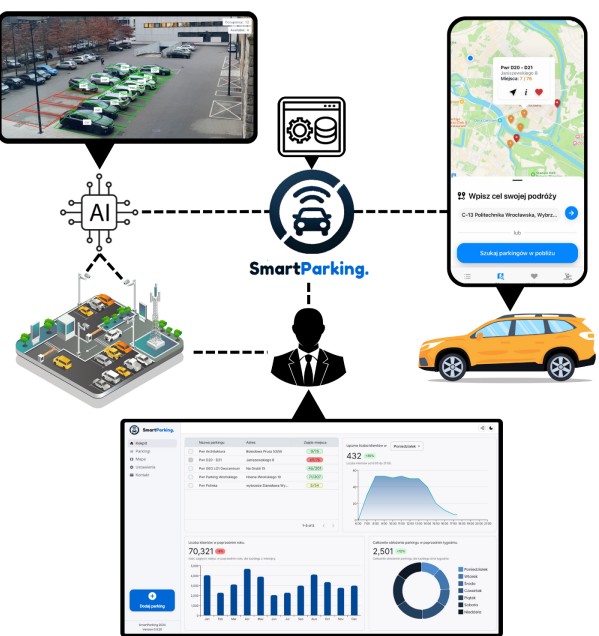

Rysunek 1: Uproszczony schemat systemu Smart Parking [4]

### 1.1 Korzyści

Unikatowa aplikacja Smart Parking ma na celu ułatwienie kierowcom dostępu do aktualnych informacji o parkingach jak i wolnych miejscach. Mogą oni w prosty i wygodny sposób skrócić czas podróży do centrów miast. Aplikacja mobilna pomoże rozważyć inne opcje transportu jak na przykład skorzystanie z Park&Ride.

Właściciele parkingów będą mieli możliwość dotarcia do większego grona odbiorców oraz analizy statystyk obłożenia swojego parkingu. Dla małych parkingów bez systemu zliczania miejsc wspierane jest rozwiązanie z analizą obrazu kamery przez model AI.

### 1.2 Cele długoterminowe

Długoterminowe cele:
- wdrożenie modelu AI do analizy miejsc parkingów
- zwiększenie liczby kierowców parkujących na mało popularnych parkingach
- zminimalizowanie problemu korków w centrach miast
- większa świadomość wydajności infrastruktury miejskiej
- umożliwienie łatwego prowadzenia statystyk zajętości parkingu nawet dla mniejszych parkingów bez systemu liczenia miejsc.

### 1.3 Zadania zrealizowane do osiągnięcia celów projektu

Prace projektowe były prowadzony w metodologii Scrum zarządzanej za pomocą oprogramowania Jira [1]. Zadania w projekcie zostały podzielone na sześć sprintów, gdzie każdy, poza ostatnim, trwał tydzień. Zostały określone główne zgłoszenia, do których przydzielono odpowiednie historyjki użytkowników.

Pierwszy sprint obejmował organizację pracy, wyboru technologii, środowiska i frameworków. Zostali wyznaczeni członkowie zespołu odpowiedzialni za konkretne części projektu, a następnie przygotowano diagramy przypadków użycia oraz wdrożenia.

Podział systemu na frontend, backend, aplikację mobilną oraz moduł AI pozwolił na rozpoczęcie pracy równolegle. Frontend realizuje wszystkie funkcje dla właścicieli parkingów umożliwiając zakładanie nowych obiektów oraz zarządzanie już istniejącymi. Dodatkowo pozwala administratorom całego systemu na organizację użytkownikami. Backend z bazą danych zarządza informacjami o parkingach oraz skupia się na zbieraniu zajętości miejsc i przesyłaniu informacji na aplikację mobilną w czasie rzeczywistym. Aplikacja mobilna realizuje funkcjonalności dla kierowców. Posiada listę z parkingami, mapę oraz możliwość nawigacji do jednego z nich.

Na koniec został wdrożony system z AI, który rozpoznaje zajęte miejsca parkingowe i wysyła informacje o nich do backendu. Przeprowadzono odpowiednie testy, na których podstawie wybrano model.

## 2 POWIĄZANE PRACE

Utworzenie systemu wynikało z analizy rynku i identyfikacji luki w dostępnych rozwiązaniach, które oferowałyby odpowiednie funkcje praktyczne. Projekt więc powstał w odpowiedzi na istniejące potrzeby oraz brak narzędzia spełniającego oczekiwania rynku.

Główną jego przewagą nad konkurencją jest obopólna korzyść ze strony klienta systemu i użytkownika końcowego aplikacji. Umożliwia on właścicielom parkingów na wygodne zbieranie i udostępnianie danych z istniejącej infrastruktury parkingowej, nawet w przypadku gdy ogranicza się ona jedynie do kamer monitorujących. Z drugiej strony użytkownik końcowy otrzymuje aktualne dane o lokalizacjach oraz zajętości miejsc parkingowych w wybranej okolicy.

Głównym konkurentem na rynku lokalnym jest system ParkSpace ECO, który udostępnia aktualnie szacowaną dostępność miejsc na parkingach miejskich w postaci trzech kolorów. Aplikacja ta ogranicza się jedynie do klientów w postaci zarządów miast oraz nie pozwala na śledzenie faktycznej liczby dostępnych miejsc w czasie rzeczywistym. Nasz system podejmuje kolejny krok w tej kwestii i pozwala miastom na integrację z istniejącą infrastrukturą parkingową. Przykładami mogą być parkingi przy Galerii Dominikańskiej oraz centrum handlowym Renoma we Wrocławiu, gdzie już istnieją systemy zliczania miejsc. Nie jest jednak możliwe uzyskanie tej informacji z poziomu smartfona przed dojazdem do celu. Smart Parking wychodząc naprzeciw oczekiwaniom użytkowników, przy możliwym wsparciu miasta, umożliwiłby rozwiązanie tego problemu. Dodatkowo wbudowany system AI pozwala na odczyt bieżącej zajętości publicznych miejsc parkingowych za pomocą zainstalowanego już monitoringu. Ponadto Smart Parking nie ogranicza się jedynie do klientów w postaci miast i oferuje te same funkcje również prywatnym właścicielom parkingów.

Innymi rozwiązaniami o charakterze konkurencyjnym są aplikacje, takie jak EasyPark oraz mobiParking. Ich funkcjonalności skupiają się głównie na opłacie za parkingi i nie są bezpośrednią konkurencją zagrażający pozytywnemu przyjęciu systemu Smart Parking.

Na rynku istnieje również konkurencja działająca poza granicami Polski. Przykładem może być firma Smart Parking Ltd. Oferuje ona zaawansowany system zarządzania całą infrastrukturą parkingową. Warto jednak napomnieć, że jej model biznesowy opiera się wyłącznie na sprzedaży produktu właścicielom parkingów. Nie bierze ona w żaden sposób pod uwagę potrzeby kierowców, którzy chcieliby poznać lokalizację i stan ich zapełnienia.

Biorąc powyższe pod uwagę, system Smart Parking powstał w odpowiedzi na brak kompleksowego rozwiązania na rynku, które łączyłoby potrzeby właścicieli parkingów z oczekiwaniami użytkowników końcowych. W przeciwieństwie do istniejących aplikacji i systemów, oferuje on integrację z infrastrukturą parkingową, aktualne dane o dostępności miejsc dla użytkowników pojazdów oraz minimalizację kosztów wdrażania dzięki wykorzystaniu istniejących kamer monitoringu.

## 3 WYNIKI

Pracę nad projektem oparto na rozwoju 4 głównych komponentów: aplikacji webowej dla właścicieli parkingów oraz władz miejskich, aplikacji mobilnej dla kierowców, rozwiązania AI do detekcji liczby wolnych miejsc oraz backendu całego systemu.

### 3.1 Funkcjonalności

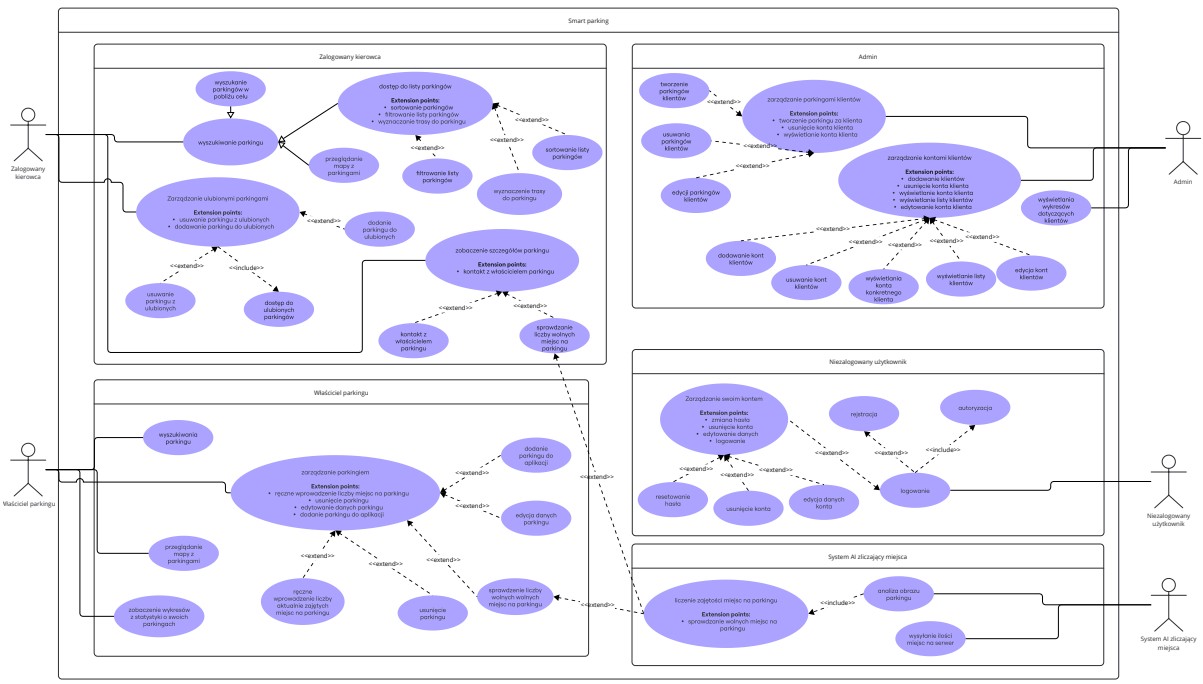

Rysunek 2: Diagram przypadków użycia

Podstawową funkcją systemu jest znajdowanie parkingów przez użytkowników. Kierowcy mogą korzystać z filtrów i opcji sortowania, co ułatwia dopasowanie wyników wyszukiwania do ich potrzeb. Dostępna jest również opcja zarządzania ulubionymi parkingami, która umożliwia szybki podgląd często używanych lokalizacji. Kierowcy mają także dostęp do mapy, na której wyświetlane są parkingi w pobliżu aktualnego położenia użytkownika lub podanego adresu. Na ekranie mapy widoczna jest lista kilku najbliższych parkingów. Użytkownicy mogą uruchomić nawigację do wybranego parkingu w aplikacji Mapy Google lub Apple Maps zarówno z poziomu listy, jak i mapy.

Niezalogowani użytkownicy mają możliwość założenia konta lub zalogowania się. W aplikacji mobilnej można założyć konto kierowcy. Konto właściciela parkingu jest przydzielane przez administratora po wykupieniu odpowiedniego pakietu.

Właściciele parkingów, jako kluczowi interesariusze systemu, mogą przeglądać dane dotyczące swoich parkingów oraz korzystać z zaawansowanych statystyk. Mają możliwość integracji istniejącego sys-

temu zliczania miejsc z systemem Smart Parking lub wdrożenia systemu AI, który zlicza miejsca na podstawie danych z kamer. Dodatkowo właściciele mają dostęp do wszystkich niezbędnych opcji zarządzania swoimi parkingami, takich jak edycja danych, usuwanie oraz dodawanie nowych parkingów.

Administratorzy systemu dysponują szerokimi możliwościami zarządzania kontami klientów i parkingami. Mogą dodawać nowe konta, edytować dane, usuwać klientów oraz zarządzać informacjami dotyczącymi parkingów, co zapewnia pełną kontrolę nad funkcjonowaniem platformy.

System uwzględnia także potrzeby użytkowników indywidualnych w zakresie zarządzania kontami. Zalogowani użytkownicy mogą edytować swoje dane, zmieniać hasła oraz, w razie potrzeby, usuwać konto. Dzięki temu mają pełną autonomię i kontrolę nad swoimi danymi osobowymi.

## 3.2 Opis techniczny – wykorzystane technologie

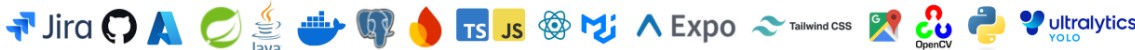

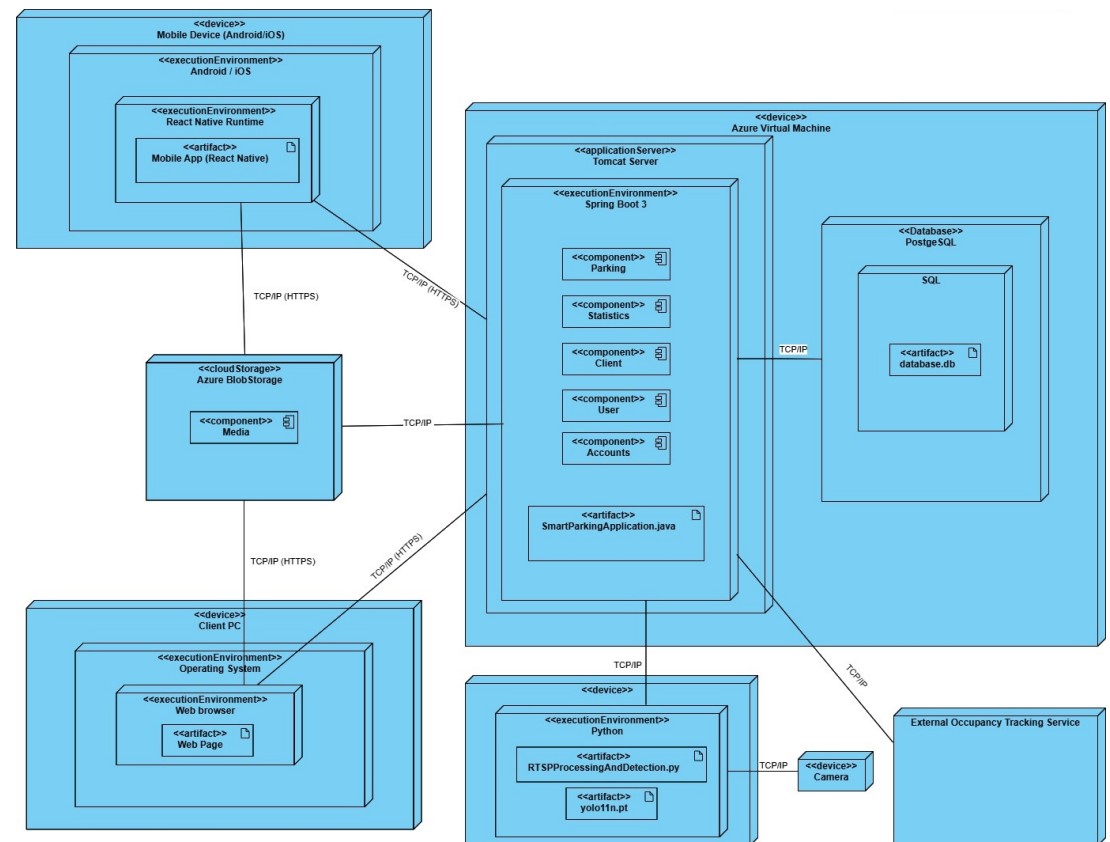

Rysunek 3: Diagram wdrożenia

### 3.2.1 Strona internetowa (Frontend)

Aplikacja webowa została napisana z wykorzystaniem biblioteki React oraz komponentów Material Design, co zapewnia spójność z estetyką opracowaną przez Google. Witryna jest responsywna, co umożliwia jej wyświetlanie zarówno na komputerach, jak i na urządzeniach mobilnych. Strona internetowa skierowana jest do właścicieli parkingów i administratora systemu. Dodatkowo, na stronie głównej dostępne są wykresy prezentujące statystyki z wybranego okresu czasu. Ponadto, udostępnia ona możliwość integracji parkingu z bieżącym systemem poprzez adres URL oraz klucz API. Klienci mogą również ręcznie wprowadzać liczbę aktualnie wolnych miejsc.

Aplikacja internetowa z uwagi na swoją złożoność implementuje kilka podstawowych wzorców. Pierwszym z nich to wzorzec Kompozyt, polegający na zdefiniowaniu podstawowych komponentów, które następnie są używane w większych komponentach – ekranach. Wprowadzony również został wzorzec warstwowy, w którym określone zostały: serwisy, komponenty, ekrany oraz najważniejsze typy. Serwisy

odpowiedzialne są za komunikację API i obsługę danych z backendu, jest to warstwa logiki biznesowej. Komponenty i Ekrany to warstwa prezentacji. Typy pełnią kluczową rolę we wspieraniu efektywnego zarządzania oraz komunikacji pomiędzy poszczególnymi warstwami aplikacji. Dodatkowo, wdrożona została centralna konfiguracja stylów oraz motywów kolorystycznych, co zapewnia spójność wizualną projektu. Rozwiązanie to doskonale integruje się z założeniami Material Design, umożliwiając szybkie i łatwe wprowadzanie zmian w motywach aplikacji.

### 3.2.2   Aplikacja Mobilna na platformy IOS oraz Android

Aplikacja mobilna została stworzona z wykorzystaniem React Native, co umożliwiło tworzenie aplikacji multiplatformowej, dzieląc jeden kod dla systemów iOS i Android. Wykorzystano język JavaScript ze względu na jego szerokie wsparcie i łatwą integrację z React Native. Do uproszczenia procesu tworzenia użyto platformy Expo, a dzięki narzędziu Expo Go możliwe było szybkie testowanie aplikacji bez konieczności budowania plików instalacyjnych.

Stylizacja interfejsu została zrealizowana przy użyciu Tailwind CSS, co pozwoliło na szybkie tworzenie nowoczesnych i responsywnych widoków. Do wyszukiwania adresów i konwersji na współrzędne geograficzne zastosowano Google Places API, zapewniające dostęp do rozbudowanej bazy danych. Obsługa map oparta została na bibliotece React Native Maps, wykorzystując Apple Maps na iOS oraz Google Maps na Androidzie. Lokalizacja użytkownika jest pobierana z funkcji natywnych telefonu, co zapewnia precyzyjne dane w czasie rzeczywistym. Autentykacja została wdrożona za pomocą Firebase z dodatkową weryfikacją po stronie backendu, co zwiększa bezpieczeństwo danych.

### 3.2.3   Backend

Backend aplikacji został zaprojektowany w Javie 21 z wykorzystaniem frameworka Spring, co zapewnia wydajność i łatwość dalszego rozwoju. Aby uprościć konfigurację wykorzystano Spring Boot. Użyte moduły Springa to: Spring Web, Spring Data JPA, Spring Security, Spring Test. W celu poprawy produktywności i czytelności kodu zastosowano Lombok do generowania powtarzalnych elementów, takich jak gettery i settery oraz MapStruct, który automatyzuje mapowanie obiektów. Komunikacja z backendem odbywa się poprzez REST API, wszystkie opakowywane są w DTO.

Do przechowywania danych została wykorzystana baza PostgreSQL. Wpływ na wybór bazy SQL miał fakt, że wszystkie dane mają stałą strukturę oraz wydajność SQL w przypadku operacji JOIN, gdzie przy odczycie parkingów brane pod uwagę są nie tylko dane samego parkingu, ale także możliwość filtrowania po ulubionych, sortowanie po lokalizacji czy też w przyszłości po ocenach. Migracje schematu bazy danych są zarządzane za pomocą Liquibase, co umożliwia łatwe i bezpieczne wdrażanie zmian struktury danych.

Autentykację użytkowników oparto o zewnętrzne rozwiązanie Firebase Authentication od Google. Pozwala ono na przesunięcie odpowiedzialności za zarządzanie kontami na zewnętrzny podmiot i w prosty sposób pozwala na logowanie się przez SSO. Przesyłane żądania na backend zawierają token JWT w nagłówku, dzięki któremu w szybki, bezstanowy i bezpieczny sposób autoryzowane są żądania.

Architektura powstała w oparciu o podejście modularne, gdzie każda domena jest odrębnym modułem, udostępniającym fasadę jako jedyny punkt dostępu dla pozostałych części systemu. Takie podejście minimalizuje zależności między modułami, zwiększa czytelność kodu i przygotowuje na przyszłe potrzeby, takie jak skalowanie i potencjalny rozwój w kierunku architektury mikroserwisowej.

### 3.2.4   System detekcji liczby wolnych miejsc

System detekcji liczby wolnych miejsc został oparty na wykorzystaniu modelu YOLOv11n z biblioteki Ultralytics. Jest to najdokładniejszy i najszybszy [2] model z serii YOLO (You Only Look Once) czyli zaawansowanych algorytmów stosowanych w wizji komputerowej. W ramach realizacji celu związanego z przetwarzaniem obrazu na parkingach wykorzystano rozwiązanie Parking Management z tej samej biblioteki [3]. Za jego pomocą przeprowadzana jest detekcja obiektów na wskazanych wcześniej obszarach parkingowych. Kod tworzony był w języku Python ze względu na jego doskonałe wsparcie przy tworzeniu rozwiązań związanych z uczeniem maszynowym.

W celu obsługi przetwarzania wideo, system wykorzystuje bibliotekę OpenCV, która odpowiada za przechwytywanie, przetwarzanie i zapisywanie klatek wideo. OpenCV działa jako pośrednik pomiędzy strumieniem wideo a modelem YOLO, dostarczając kolejne klatki do analizy i obsługując ich wizualizację w postaci przetworzonych obrazów z naniesionymi wynikami detekcji. Na podstawie tej analizy system określa liczbę dostępnych i zajętych miejsc, a następnie przesyła je na odpowiedni endpoint do backendu.

### 3.2.5 Wdrożenie

Backend i frontend aplikacji zostały skonteneryzowane przy użyciu Dockera i wdrożone na wirtualną maszynę w chmurze Azure. Ponadto wykorzystano Azure Blob Storage do hostowania zdjęć z aplikacji.

## 3.3 Testy

### 3.3.1 Testy akceptacyjne i inne

| Przypadek użycia | Opis testu | Wynik |
|---|---|---|
| Wyszukanie parkingów w pobliżu celu | - Uruchomienie aplikacji mobilnej
- Zalogowanie się ręcznie lub automatycznie
- Wpisanie adresu w miejsce "Gdzie chciałbyś dojechać?"
- Wybranie pozycji z listy
- Naciśnięcie niebieskiej strzałki
- Przeglądanie listy parkingów obok wybranego miejsca z widokiem na mapę | Pozytywny |
| Wyznaczenie trasy do parkingu wyszukanego na liście w aplikacji mobilnej | - Uruchomienie aplikacji mobilnej
- Zalogowanie się ręcznie lub automatycznie
- Kliknięcie ikonki listy na pasku dolnym
- Kliknięcie w pole "Wyszukaj parking"
- Wpisanie frazy wyszukiwania
- Naciśnięcie lupy
- Naciśnięcie ikony ze strzałką nawigacji przy parkingu do którego należy dojechać
- Automatyczne otwarcie aplikacji z nawigacją | Pozytywny |
| Rejestracja użytkownika na aplikacji mobilnej | - Uruchomienie aplikacji mobilnej
- Naciśnięcie tekstu "Załóż konto"
- Podanie nazwy użytkownika, emaila i hasła
- Naciśnięcie przycisku "Zarejestruj się"
- Kliknięcie w link aktywujący konto, który przyszedł na podanego maila
- Zalogowanie się do aplikacji za pomocą podanych danych | Pozytywny |
| Dodanie parkingu | - Wejście na stronę internetową
- Kliknięcie 'Dodaj parking' na bocznym pasku
- Dodanie zdjęcia
- Wypełnienie danych
- Modyfikacja pinezki na mapie
- Kliknięcie przycisku zapisz | Pozytywny |

Tabela 1: Lista przykładowych testów akceptacyjnych

W powyższej tabeli zostały pokazane przykładowe testy akceptacyjne. Łącznie testów akceptacyjnych było 44 i pokrywały one wszystkie przypadki użycia. Każdy z nich zakończył się sukcesem. Ponadto w celu weryfikacji kodu przeprowadzono testy jednostkowe, integracyjne, automatyczne oraz manualne.

### 3.3.2 Wyniki modelu AI

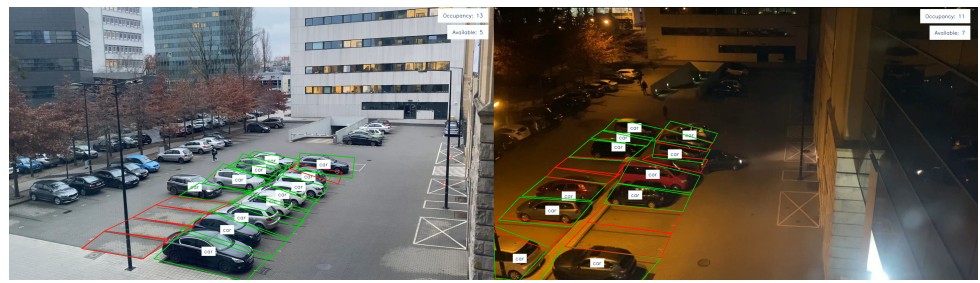

Rysunek 4: Detekcja liczby wolnych miejsc na parkingu Politechniki Wrocławskiej

Model detekcji liczby wolnych miejsc został przetestowany na parkingach Politechniki Wrocławskiej zarówno w ciągu dnia jak i wieczorem dla porównania wyników w różnym oświetleniu. Metryki oceny jakości modelu takie jak dokładność, precyzja, czułość i swoistość wyliczone zostały na podstawie macierzy pomyłek, gdzie klasą pozytywną były miejsca wolne, a negatywną zajęte. Ponadto zbadano czas inferencji jednej klatki. Na poniższej tabeli przedstawiono porównanie wyników modelu uśrednione na 6 klatkach przedstawiających różne konfiguracje aut. Ujęcia pochodzą z filmów nagranych na tym samym parkingu w dwóch różnych porach dnia. Wyliczeniom podległo 18 miejsc parkingowych.

| Nagranie | Dokładność | Precyzja | Czułość | Swoistość | Czas CPU (ms) | Czas GPU (ms) |
|----------|------------|----------|---------|-----------|---------------|---------------|
| Wieczorem | 88% | 80.3% | 86.5% | 88.8% | 144.22 | 34.2 |
| W dzień | 88.9% | 78% | 90.5% | 87.9% | 172.29 | 49.8 |

Tabela 2: Wyniki modelu o różnych porach dnia.

### 3.3.3 Zrzuty ekranu

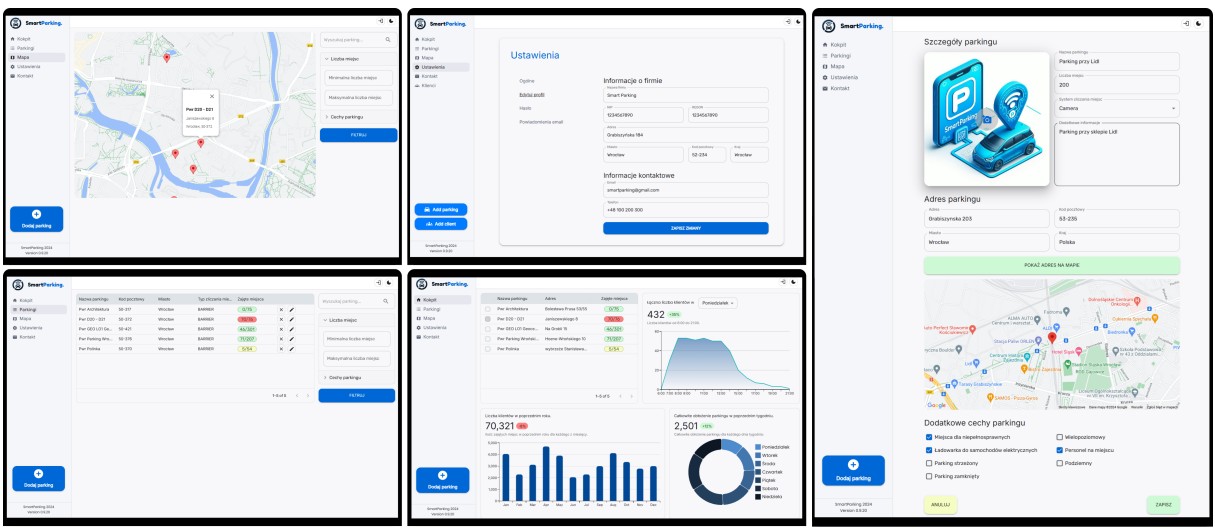

Rysunek 5: Interfejsy aplikacji webowej

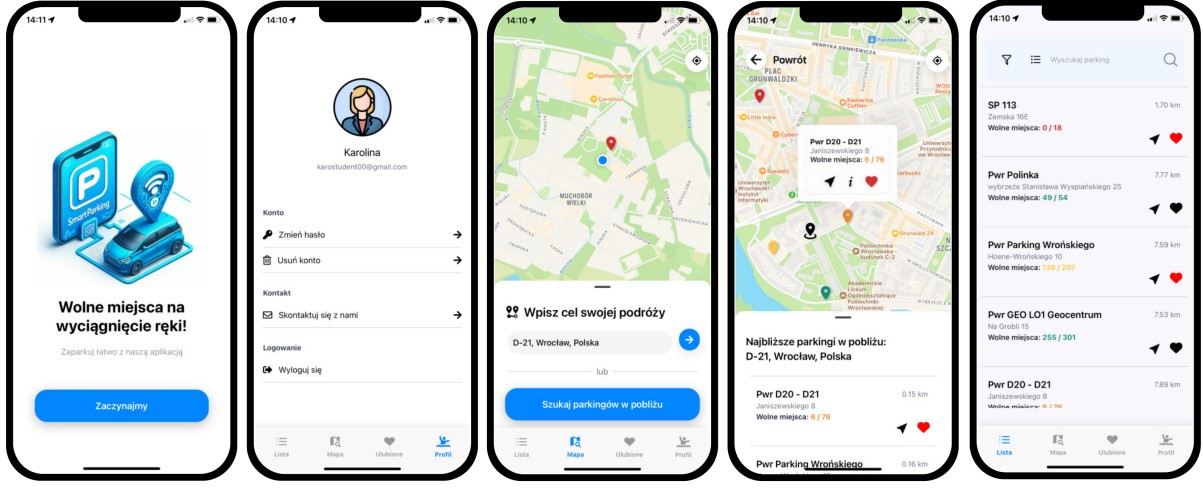

Rysunek 6: Interfejsy aplikacji mobilnej

### 3.4 Wartość dodana

**Dla właścicieli parkingów:**

- zwiększenie widoczności parkingu - dzięki integracji systemu z aplikacją, właściciele parkingów, mogą łatwo dotrzeć do większej liczby potencjalnych klientów.

- lepsze zarządzanie obłożeniem - system umożliwia właścicielom parkingów analizowanie obłożenia, miejsc parkingowych w czasie rzeczywistym. Dzięki temu mogą oni dostosowywać ceny oraz dostępność miejsc w zależności od pory dnia, lokalizacji czy specyficznych potrzeb klientów,

- potencjalne przyszłe inwestycje - zebrane dane o dostępności parkingów mogą być wykorzystane przez władze miejskie do lepszego planowania infrastruktury parkingowej.

**Dla kierowców:**

- oszczędność czasu - jedną z najważniejszych zalet dla kierowców jest możliwość szybkiego znalezienia dostępnego miejsca parkingowego. Aplikacja umożliwia wyszukiwanie parkingów w oparciu o lokalizację, dostępność miejsc i preferencje,

- intuicyjny interfejs użytkownika - aplikacja jest zaprojektowana w sposób, który zapewnia łatwą i szybką nawigację, co pozwala kierowcom na błyskawiczne znalezienie informacji.

## 4 PODSUMOWANIE

Podsumowując, system Smart Parking to uniwersalne rozwiązanie, które wspiera zarówno właścicieli parkingów, jak i kierowców.

Właściciele korzystający z tego systemu mają możliwość dodatkowej reklamy. Ponadto zyskują oni wgląd do statystyk o obłożeniu parkingów. Jest to kolejna wartość dodana jako, iż na ich podstawie mogą na przykład dostosowywać ceny w zależności od pory dnia. Z kolei władze miast mogą wykorzystać te statystyki do wskazania obszarów o niewystarczającej liczbie miejsc parkingowych oraz tych, gdzie miejsc jest nadmiar. Dzięki temu mogą efektywnie planować przyszłe inwestycje. Kolejną interesującą cechą systemu jest rozwiązanie AI. Umożliwia w wygodny i niewymagający dużych nakładów finansowych sposób zliczanie miejsc jedynie na podstawie nagrania z monitoringu.

Kierowcy natomiast zyskują dostęp do informacji o wolnych miejscach parkingowych, co pozwala zaoszczędzić im czas i nerwy podczas szukania miejsca. Ta funkcjonalność jest filarem aplikacji i czyni ją jedyną taką na rynku. Całość opakowana w intuicyjną mobilną wersję z pewnością sprosta oczekiwaniom użytkowników.

Rozwiązanie to z pewnością nie wyczerpuje tematu i wciąż może zostać poszerzone o wiele pożądanych funkcjonalności. Są to między innymi:
- system powiadomień o braku wolnych miejsc na parkingu do którego jedzie użytkownik,
- możliwość odpłatnego promowania parkingów i częstsze wyświetlanie ich użytkownikom,
- system ocen parkingów,
- możliwość płatności za parkowanie bezpośrednio w aplikacji,
- dla parkingów wspomaganych systemem AI wskazywanie lokalizacji wolnego miejsca parkingowego.
Dzięki innowacyjności i praktycznym rozwiązaniom Smart Parking ma szansę podbić rynek rozwiązań parkingowych i zdobyć zaufanie użytkowników.

## LITERATURA

[1] Atlassian. *Dokumentacja Jira*, 2024. Dostęp: 30 listopada 2024.

[2] Nidhal Jegham, Chan Young Koh, Marwan Abdelatti, and Abdeltawab Hendawi. Evaluating the evolution of yolo (you only look once) models: A comprehensive benchmark study of yolo11 and its predecessors, 2024.

[3] Ultralytics. *Parking Management Guide*, 2024. Dostęp: 30 listopada 2024.

[4] Vecteezy.com. *https://www.vecteezy.com/free-vector*.
