# OpenReview forum: "Smart Parking"
_pwr.edu.pl/Wrocław_University_of_Science_and_Technology/2024/ZPI_Day — Wrocław University of Science and Technology 2024 ZPI Day Submission_

### Official Review · Reviewer_MHxU · 2024-12-04
**Problem wyszukiwania miejsc parkingowych - w dobie ich niedoboru czy wręcz ich braku - jest problemem ważnym z praktycznego punktu widzenia.**

**Confidence:** 4
**Significance Of Results:** 4
**Overall Quality:** 5

**Compliance With Template:**

5: Very High Quality – The article contains all the required sections, which are written in a very detailed, clear, and error-free manner. The structure is professional and meets expectations, and the content adheres to the highest substantive and formal standards.

**Description Of Results:**

4: High Quality – The results are described in detail and supported by usage examples or evaluations. The description is reliable but may lack full depth of analysis.

**Feedback On Consistency:**

Przedstawione rozwiązanie stanowi kompleksowe wykorzystanie i synergetyczne połączenie istniejących rozwiązań, gdzie na jednym z końców wykorzystanych technologii stoi szeroko stosowana biblioteka OpenCV wykorzystywana w wielu projektach - tak naukowych, jak i komercyjnych. Opis projektu jest logiczny i spójny. Wykorzystywany język jest prawidłowy, a przekaz tekstu jasny i klarowny.

**Potential For Development:**

Projekt - po uwzględnieniu obostrzeń licencyjnych wszystkich zastosowanych komponentów mógłby zostać wdrożony komercyjnie. Wskazane byłoby jednak przeprowadzenie badań czytelności danych uzyskiwanych ze strumienia wideo w przypadku słabszej jakości strumienia (gorsza widoczność podczas deszczu, mgły).

**Project Nature Evaluation:**

Projekt ma charakter inżynierski i spełnia wszystkie założenia stawiane przed tego typu zadaniami. Wykorzystane narzędzia i metody są dobrane i zastosowane w przemyślany i prawidłowy sposób.

**Technical Language Precision:**

5: Very High Quality – The language is entirely appropriate for a technical report. All terms are used correctly and precisely, and the style is professional, clear, and coherent, without any errors or ambiguities.

---

### Official Review · Reviewer_anX1 · 2024-12-06
**A review of a novel parking system using AI solutions.**

**Confidence:** 5
**Significance Of Results:** 5
**Overall Quality:** 4

**Compliance With Template:**

5: Very High Quality – The article contains all the required sections, which are written in a very detailed, clear, and error-free manner. The structure is professional and meets expectations, and the content adheres to the highest substantive and formal standards.

**Description Of Results:**

4: High Quality – The results are described in detail and supported by usage examples or evaluations. The description is reliable but may lack full depth of analysis.

**Feedback On Consistency:**

The problem description consistency is high enough. Authors present the project technical details, the system architecture and the project objectives. The commercialization potential is very high. The object detection model applied provides very interesting and flexible solution for parking management.

**Potential For Development:**

The authors show great potential for system development and commercial opportunities.

**Project Nature Evaluation:**

This is an engineering work. Technical details are provided as well.

**Technical Language Precision:**

4: High Quality – The language is appropriate for a technical report. Terminology is used correctly, and statements are precise, with only minor shortcomings that do not affect the overall clarity.

---

### Official Review · Reviewer_5jHb · 2024-12-07
**Recenzja Smart Parking**

**Confidence:** 4
**Significance Of Results:** 4
**Overall Quality:** 5

**Compliance With Template:**

4: High Quality – The article contains all the required sections, which are well-written and substantively correct, although minor errors or shortcomings may be present. The overall structure is clear and coherent.

**Description Of Results:**

5: Very High Quality – The results are described in detail, clearly and comprehensively, supported by thorough evaluation, analysis, and convincing usage examples. The description meets the highest substantive standards.

**Feedback On Consistency:**

Analiza problemu, cel i zakres funkcjonalny sposób został przedstawiony w zwięzły i jasny sposób. Prezentacja wyników jest przedstawiona w kontekście zdefiniowanych problemów. Podsumowanie i proponowana dalsza praca na projektem wydaje się być logicznym następstwem uzyskanych rezultatów.

**Potential For Development:**

Wysoki potencjał rozwoju, czego autorzy dają wyraz w sekcji podsumowanie,w szczególności w kierunku wykorzystania AI. Zdecydowanie projekt do wdrożenie komercyjnego.

**Project Nature Evaluation:**

Proces implementacji, oraz sama implementacja wydają się być na wysokim poziomie. Wybrane rozwiązania technicznie są wybrana świadomie, bez zbędnego nadmiaru i w oczywisty sposób realizują potrzeby projektu. Dokładnie opisana architektura i projekt i udokumentowana modelami. Uzasadniona integracja z AI.

**Technical Language Precision:**

5: Very High Quality – The language is entirely appropriate for a technical report. All terms are used correctly and precisely, and the style is professional, clear, and coherent, without any errors or ambiguities.

---

### Decision · Program_Chairs · 2024-12-10

Accept (Oral)